# Choosing from an Optimal Number of Options Makes Curry and Tea More Palatable

**DOI:** 10.3390/foods8050145

**Published:** 2019-04-29

**Authors:** Takuya Onuma, Nobuyuki Sakai

**Affiliations:** 1Department of Management and Business, Faculty of Humanity-oriented Science and Engineering, Kindai University, Kayanomori 11-6, Iizuka, Fukuoka 820-8555, Japan; t.onuma@fuk.kindai.ac.jp; 2Department of Psychology, Graduate School of Arts and Letters, Tohoku University, Kawauchi 27-1, Aoba-ku, Sendai 980-8576, Japan; 3Multimodal Cognitive System Laboratory, Research Institute of Electrical Communication, Tohoku University, Katahira 2-1-1, Aoba-ku, Sendai 980-8577, Japan; 4Advanced Institute for Yotta Informatics, Tohoku University, Kawauchi 27-1, Aoba-ku, Sendai 980-8576, Japan

**Keywords:** choice, palatability perception, the number of options, curry, tea

## Abstract

Previous studies have shown that affording people choice increases their satisfaction with subsequent experiences: the choice effect. However, it remains unclear whether the choice effect occurs in the hedonic response to foods and beverages. Thus, the present study aimed to demonstrate the choice effect on the palatability perception. Ready-to-serve curries and tea were presented as options in Experiment 1 and Experiment 2, respectively. Experiment 1 failed to demonstrate significant differences among palatability ratings for a curry chosen by participants and for a curry chosen by the experimenter. However, Experiment 2 demonstrated that participants perceived a tea chosen by themselves as more palatable than another tea chosen by the experimenter, regardless of the fact that the two cups of tea were identical. Intriguingly, the effect was obtained only when the number of options was neither too small nor too big. These results indicate that the exercise of choice from an optimal number of options, even when the choice is ostensible and illusory, makes people perceive their chosen foods and beverages as being more palatable. Some implications for the domain of food business are also discussed.

## 1. Introduction

It is generally believed that our hedonic responses to foods and beverages are simply determined by their physical and chemical properties. However, a growing body of studies has demonstrated that hedonic responses to foods and beverages are significantly affected by contextual factors such as the order of tasting [1,2], plateware and cutlery [3,4], background music [5,6], prices [7], and brands [8,9,10]. These studies indicate that we should pay attention not only to the physical and chemical properties of foods and beverages, but also to the effect of contextual factors in food perception to fully understand consumers’ hedonic responses.

When the hedonic responses to foods and beverages are investigated by researchers, several different aspects, such as palatability, pleasantness, liking, or preference are discussed. Generally, palatability and pleasantness are thought to be hedonic experience of the sensory aspects of foods and beverages, while liking and preference are thought to be cognitive judgement or resultant selection behavior [2,11,12]. The present study focused on the palatability of foods and beverages, and aimed to examine the effect of choice on palatability perception.

In our everyday life, we, as consumers, often struggle to choose the best item from a number of items in supermarkets or online marketplaces. Choice is customarily believed to be an opportunity to match a preference with the available options. Contrary to this belief, recent psychological studies have demonstrated that choice affects the response to its outcome; people who were afforded a chance to choose were more satisfied with their chosen option or performed better in their chosen activity or task compared to their counterparts who were not afforded the chance to choose but were simply given the same option or activity [13,14,15,16]. If this effect, called *the choice effect* in the present study, is applicable to the food domain, appropriately affording people choice can lead them to perceiving a food or beverage as being more palatable. If the choice effect occurs in food perception, people who are afforded the chance to choose would perceive their chosen food as being more palatable than others who are merely given the food. Moreover, people would perceive a food chosen by themselves as more palatable than another food chosen by others, even when the foods are the same.

This study aims to demonstrate the choice effect on the palatability perception of foods and beverages. The present study consists of two experiments. A variety of ready-to-serve curries and tea were presented as options in Experiment 1 and Experiment 2, respectively. Participants were asked to choose one item to taste and evaluated its palatability. In Experiment 1, the choice effect was examined in terms of expectation before tasting. In Experiment 2, the choice effect was examined in terms of the number of options presented.

## 2. Experiment 1

### 2.1. Overview

This experiment aimed to demonstrate the choice effect on palatability perception of the chosen curry and to examine its relationship with expectation. In our daily lives, people normally choose one item from various options because they expect that the item will provide them with the greatest experience or utility among the available options; choice often concurs with greater expectation. Since expectations about foods or beverages are known to affect the actual experience of foods or beverages [17,18,19], the choice effect can simply be driven by the greater expectation of the option. Thus, this possibility was examined in this experiment.

In this experiment, nine kinds of ready-to-serve curries were presented as options. Participants were asked to rank the curries based on their expected palatability (i.e., first to ninth), and choose one curry they wanted to taste from three curries ranked medium (i.e., fourth to sixth). Participants were randomly allocated into one of the three experimental conditions: the mid-choice condition, the high-given condition, or the low-given condition. In the mid-choice condition, participants were told that they were going to be served the curry they had chosen. In the high- and low-given conditions, on the other hand, participants were told that they were going to be served a higher- or lower-ranked curry that the experimenter had chosen. Regardless of what the participants chose or believed to taste, the curry that was actually served and tasted was identical across the conditions, which enabled us to control the choice outcome and examine the choice effect more directly [20,21]. If the choice effect is simply driven by greater expectation, the participants in the high-given condition would perceive the curry as being more palatable than those in the mid- and low-given conditions. On the other hand, if the choice itself plays a key role in the choice effect, participants in the mid-choice condition, whose choice was accepted, would perceive the curry as being more palatable than those in the high- and low-given conditions.

### 2.2. Materials and Methods

#### 2.2.1. Participants

Thirty-one university students (17 men and 14 women; *M_age_* = 20.7 years) participated in this experiment. Participants were randomly allocated into the mid-choice (*n* = 10), high-given (*n* = 10), or the low-given condition (*n* = 11).

Verbal and written explanations about the experiment were provided to the participants and written informed consent signed by the participants was obtained before the experiment. After the experiment, participants were debriefed of the true purpose of the experiment, and again gave written informed consent. This experiment was conducted according to the Declaration of Helsinki for Research involving Human Subjects, and received approval from the Ethics Committee of the Graduate School of Arts and Letters, Tohoku University.

#### 2.2.2. Stimuli

Nine kinds of ready-to-serve curries available on the Japanese market were used as visual stimuli: *Curry Marche*, *Java Curry*, *Uruoi Recipe Yawaraka Beef Curry*, *Delhi Premium Recipe Beef Masala Curry*, *Torouma Gyu Kakuni Curry* (House Foods Corporation, Tokyo, Japan), *Ginza Curry* (Meiji Corporation, Tokyo, Japan), *Indian Curry Beef Spicy* (Nakamuraya Corporation, Tokyo, Japan), *Bon Curry* (Otsuka Foods Company, Osaka, Japan), and *Curry Youbi* (S&B Foods Inc., Tokyo, Japan). In the choice session, these visual stimuli were presented as options and participants were asked to choose one curry to taste (Figure 1).

In the tasting session, either of the two types of curries, Curry A (*Curry-ya Curry*, House Foods Corporation, Tokyo, Japan) or Curry B (*Curry Shokunin*, Ezaki Glico Corporation, Osaka, Japan), was presented. This aimed to examine whether the possible effect would be restricted to a certain type of curry. The identity of the presented curry was counter-balanced across the participants. The curry was prepared and delivered to participants just before tasting (20 g of curry sauce with 50 g cooked rice).

#### 2.2.3. Procedure

The experimental procedure is summarized in Figure 2. At the beginning, participants were given a cover story: the experiment aimed to investigate young Japanese people’s attitude and preference for curry. After a brief explanation of the procedure was given, in the choice session, all participants were presented with the options. Participants were asked to rank the curries from first to ninth based on their expected palatability. Groups of three curries ranked first to third, fourth to sixth, and seventh to ninth were used as the *high-ranked curries*, the *medium-ranked curries*, and the *low-ranked curries*, respectively. Participants were then asked which one of the *medium-ranked curries* they wanted to taste.

Once participants indicated their choice, participants in the mid-choice condition were told that they were going to be served the curry they had chosen from the *medium-ranked curries*. On the other hand, participants in the other two conditions received an apology from the experimenter and were told that the curry they had chosen was temporally out of stock. Based on this cover story, participants in the high-given condition were asked to taste a curry that the experimenter had randomly chosen from the *high-ranked curries*. In the same manner, participants in the low-given condition were asked to taste a curry randomly chosen from the *low-ranked curries*.

After the choice session, the experimenter started to prepare the curry behind a partition. During the preparation, participants were presented with the package of the curry they were going to taste, and asked to expect its flavor and to evaluate how palatable the curry seemed (i.e., the expected palatability) on a 55-mm visual analog scale (VAS). The scale was anchored “seems not palatable at all” and “seems extremely palatable” at the left and right ends of the scales.

Participants were then presented with the curry, and asked to taste and evaluate how palatable the curry was (i.e., perceived palatability) on the 55-mm VAS. The scale was anchored “absolutely not palatable” and “extremely palatable” at the left and right ends of the scales. Regardless of what participants believed to taste, in actuality, the curry presented in this tasting session was determined beforehand: Curry A or Curry B (counter-balanced across participants).

#### 2.2.4. Data Analysis

To obtain the expected palatability rating and the perceived palatability rating, the length (mm) from the left edge of the VAS to a mark participants had made was measured for each evaluation and converted into rating ranging from 0 to 100.

Statistical analyses were performed by the software R (version 3.5.0, The R Foundation for Statistical Computing Platform, Vienna, Austria). Since the sample size in this experiment was small, non-parametric statistical analyses were conducted. To assess the differences among the experimental conditions, Steel-Dwass test, known as a non-parametric equivalent of Tukey-Kramer test, was used as *a priori* multiple comparison of the expected palatability rating and the perceived palatability rating (the function *pSDCFlig* in the package *NSM3* in R). To examine the relationship between the expected palatability rating and the perceived palatability rating, Spearman’s rank correlation coefficient *ρ* was calculated based on the global data as well as the subsets of the experimental conditions (the function *cor.test* in R). Probability values of less than 0.05 (*p* < 0.05) were considered statistically significant.

### 2.3. Results

The results for the expected palatability rating are shown in Figure 3A. The expected palatability ratings tended to be high for the high-given condition, and low for the low-given condition. A Steel-Dwass test confirmed this observation: the high-given condition significantly differed from the low-given condition (*W* = −3.34, *p* = 0.0475). However, the mid-choice condition did not significantly differ from the high-given (*W* = −2.79, *p* = 0.12) or the low-given conditions (*W* = 1.60, *p* = 0.50).

The results for the perceived palatability rating of the sampled curries are shown in Figure 3B. Comparing the perceived palatability ratings among the experimental conditions, the ratings tended to be high for the mid-choice condition where the sampled curry was Curry B, whereas the ratings tended to be low for the mid-choice condition where the sampled curry was Curry A. However, Steel-Dwass tests found there was no significant differences among the experimental condition both for Curry A and Curry B (*p*s > 0.10).

Spearman’s rank correlation coefficients between the indices were calculated based on the global data, as well as the subsets of the experimental conditions (Table 1). However, no significant correlation between the indices was found.

### 2.4. Discussion

This experiment aimed to demonstrate the choice effect on the palatability perception of curry and examine its potential relationship with expectation. However, this experiment failed to demonstrate both the significant choice effect and its relationship with expectation.

The expected palatability rating for the high-given condition was significantly higher than that for the low-given condition. The rating was intermediate for the mid-choice condition. This result indicates that the experimental manipulation of expectation was successful.

We predicted that, if the choice effect is simply driven by greater expectation of the item, participants in the high-given condition would perceive the curry as being more palatable than those in the mid-choice and low-given conditions. However, the perceived palatability ratings were not significantly different among the conditions. Furthermore, there were no significant correlations between the expected palatability rating and the perceived palatability rating. These results indicate that expectation had little or no effect on palatability perception in this experiment.

There was a tendency that the perceived palatability rating for the mid-choice condition was higher than those for the high- and low-given conditions when the sampled curry was Curry B. This tendency was consistent with another prediction that, if the choice itself plays a key role in the choice effect, participants in the mid-choice condition whose choice was accepted would perceive the curry as being the most palatable.

Since the sensory characteristics of the sampled curries were not well investigated in the present study, it is difficult to fully interpret why the tendency was obtained only when a certain curry was sampled. One possible explanation for the ambiguous result is that the choice effect was attenuated by certain factors. For instance, in this experiment, participants were presented with nine curries. The number of options had been determined based on previous studies, suggesting that eight to ten options are ideal to increase consumers’ satisfaction with their choice [20,22,23]. However, participants in this experiment were asked to rank the nine curries, and then they were asked to choose one curry from a group of three curries (i.e., the *medium-ranked curries*). This instruction may have made participants perceive the number of options as three rather than nine, which is thought of as too small a number of options to increase consumers’ satisfaction [20,22,23]. Therefore, it was predicted that if the number of options was well manipulated and controlled, the choice effect on palatability perception would be clearly obtained. Thus, we examined this possibility in Experiment 2.

## 3. Experiment 2 

### 3.1. Overview

This experiment aimed to further examine the hypothesis that choice plays a key role in the choice effect, by manipulating and controlling the number of options. In addition, whereas Experiment 1 examined the choice effect in the between-participant design, this experiment examined it in the within-participant design, which aimed to exclude participants’ individual differences of tea preference.

In this experiment, several kinds of tea were presented as options. The number of options varied among experimental conditions: three (small), nine (medium), and twelve (large) options. Participants were asked to choose a preferable tea bag from the options. Participants were told that they were going to taste and evaluate the tea they had chosen (i.e., chosen tea) as well as another tea that the experimenter had chosen at random (i.e., given tea). However, regardless of what the participants chose or believed to taste, the two cups of tea that were served were identical throughout the experiment; all participants tasted and evaluated the same tea twice. This manipulation, as in Experiment 1, aimed to control the choice outcome and examine the choice effect more directly [20,21]. It was hypothesized that the participants would perceive the chosen tea as more palatable than the given tea, and also that the choice effect would be obtained only when the number of options was nine, but not three or twelve.

It has been suggested that consumers’ satisfaction with their choice increases when the sense of self-determination of their choice is stronger, and also that the sense of self-determination increases as a function of the perceived variety in the option assortment [21]. Therefore, to confirm this relationship, participants in this experiment were also asked to evaluate the perceived variety in the assortment and their sense of self-determination. 

### 3.2. Materials and Methods

#### 3.2.1. Participants

Fifty university students (26 men and 24 women; *M_age_* = 19.5 years) participated in this experiment. Participants were randomly allocated into the 3-option condition (*n* = 18), the 9-option condition (*n* = 17), or the 12-option condition (*n* = 15).

Verbal and written explanations about the experiment were provided to the participants and written informed consent signed by the participants was obtained before the experiment. After the experiment, participants were debriefed of the true purpose of the experiment, and again provided their written informed consent. This experiment was conducted according to the Declaration of Helsinki for Research involving Human Subjects and received approval from the Ethics Committee of the Faculty of Humanity-oriented Science and Engineering, Kindai University, Japan.

#### 3.2.2. Stimuli

Tea bags (DEAN & DELUCA, NY, United States) were used as visual stimuli. In the 3-option condition, three kinds of tea bags were used: *Dean & Deluca Blend*, *Earl Grey Extra*, and *Darjeeling*. In the 9-option condition, nine kinds of tea bags were used: *Dean & Deluca Blend*, *Earl Grey Extra*, *Darjeeling*, *Rooibos & Rose*, *Elder Flower & Chamomile*, *Moroccan Mint*, *Ginger & Lemon Myrtle*, *Apple*, and *Caramel*. In the 12-option condition, 12 kinds of tea bags were used: *Dean & Deluca Blend*, *Earl Grey Extra*, *Darjeeling*, *Rooibos & Rose*, *Elder Flower & Chamomile*, *Moroccan Mint*, *Ginger & Lemon Myrtle*, *Apple*, *Caramel*, *Holiday*, *Breakfast*, and *Decaf Earl Grey*. In the choice session, the visual stimuli were presented as options and participants were asked to choose one tea bag to taste (Figure 4).

In the tasting session, an iced straight tea with low sugar, *Koucha-no-Jikan* (UCC, Kobe, Japan), was presented twice. The tea was stored in a refrigerator and prepared just before tasting. The tea (100 mL) was delivered to the participants in a clear cup.

#### 3.2.3. Procedure

The experimental procedure is summarized in Figure 5. At the beginning, participants were given a cover story: the experiment was aimed to investigate young Japanese people’s attitude and preference for tea. After a brief explanation of the procedure, in the choice session, the participants were presented with the options. The number of options varied among the conditions: three, nine, or twelve.

Participants were then asked to evaluate how much of a sense of variety they perceived in the assortment (i.e., perceived variety in the assortment). This evaluation was conducted, based on previous studies [21,24], by asking participants “How different are the tea options from each other” and “How similar are the tea options to each other? (reversed item)” on a 7-point Likert scale (1 = not at all, 7 = very much).

After that, participants were asked which tea they wanted to taste. Once participants indicated their choice, they were asked to evaluate how much they felt that the choice was based on their own will (i.e., sense of self-determination). This evaluation was conducted, based on previous studies [21,25], by asking participants to state how true the following statements were for them on a 7-point Likert scale (1 = not at all true, 7 = very true): “I selected this particular tea because I wanted to” and “I selected this particular tea because I had no choice (reversed item).” Participants were then told that they were going to taste the tea they had chosen (i.e., chosen tea) *as well as* another tea that was randomly chosen by the experimenter (i.e., given tea). 

The experimenter prepared the tea in another room, and then participants were sequentially and separately presented with the chosen tea and the given tea. The order of presentation was counter-balanced across the participants. Each tasting session was separated by about three minutes. Regardless of what participants believed they tasted, in actuality, the two cups of tea were identical. To make participants believe that they were tasting two different teas, tea bags of the chosen tea and the given tea were presented together with the cups of tea. Participants were asked to taste and evaluate the palatability of the tea (i.e., perceived palatability) on the 100-mm VAS anchored “absolutely not palatable” and “extremely palatable” at the left and right ends of the scales.

After the tasting session, participants were asked to verbally report anything special they had felt or noticed during the experiment. Participants were then informed the true purpose of the experiment.

#### 3.2.4. Data Analysis

Some participants (*n* = 5) reported that they had noticed the chosen and the given tea being identical. They might have also noticed the true purpose of this experiment, which could have altered their response and rating. Therefore, one participant in the 3-option condition, three participants in the 9-option condition, and one participant in the 12-option condition were excluded from the following analyses.

To obtain the perceived palatability rating, the length (mm) from the left edge of the VAS to a mark participants had made was measured for each evaluation. In addition, to directly examine the choice effect on palatability perception, the palatability rating for the given tea was subtracted from that for the chosen tea for each participant, and it was used as a choice effect score (i.e., a positive value means that the chosen tea was evaluated as being more palatable than the given tea).

The perceived variety in the assortment was measured by the two different items. Since internal reliability of the two items was high (Cronbach’s α = 0.808), scores for the first item (“How different are the tea options from each other?”) and reverse scores for the second item (“How similar are the tea options to each other?”) were averaged to create an index of perceived variety.

The sense of self-determination was also measured by the two different items, but internal reliability was not high (Cronbach’s α = 0.492). Therefore, scores for the two items (“I selected this particular tea because I wanted to” and “I selected this particular tea because I had no choice”) were individually analyzed as indices of the sense of self-determination.

Since the sample size in this experiment was small, non-parametric statistical analyses were conducted. To assess the differences among the experimental conditions, as in Experiment 1, Steel-Dwass test was used as *a priori* multiple comparison of the choice effect score, the perceived variety in the assortment, and the sense of self-determination (the function *pSDCFlig* in the package *NSM3* in R). To examine whether the choice effect scores for each condition significantly differed from zero, one-sample Wilcoxon signed rank test against zero was also conducted (the function *wilcox.test* in the package *ggpubr* in R). Probability values of less than 0.05 (*p* < 0.05) were considered statistically significant.

### 3.3. Results

The results for the choice effect score among the experimental conditions are shown in Figure 6. The scores tended to be positive for the 9-option condition, whereas the scores were almost zero for the 3-option and the 12-option conditions. A Steel-Dwass test confirmed this observation: the 9-option condition significantly differed from the 3-option condition (*W* = 3.40, *p* = 0.0428), though the 9-option did not significantly differ from the 12-option condition (*W* = 2.54, *p* = 0.17). There was no significant difference between the 12-option and the 3-option conditions (*W* = 0.06, *p* = 0.99). Moreover, one-sample Wilcoxon signed rank tests revealed that the median for the 9-option condition was significantly higher than zero (*p* = 0.01915), but that for the 3-option (*p* = 0.48) and 12-option conditions (*p* = 0.78) did not differ from zero.

The results for the perceived variety in the assortment among the experimental conditions are shown in Figure 7A. The scores tended to be low for the 3-option condition, and high for the 9-option condition. However, Steel-Dwass tests found there was no significant differences among the conditions (*p*s > 0.10).

The results for the sense of self-determination among the experimental conditions are shown in Figure 7B,C. The scores for the item “I selected this particular tea because I wanted to” tended to increase as a function of the number of options, and the scores for the item “I selected this particular tea because I had no choice” tended to decrease as a function of the number of options. However, Steel-Dwass tests found there was no significant differences among the conditions for both items (*p*s > 0.10).

### 3.4. Discussion

This experiment aimed to examine the hypothesis that participants would perceive the chosen tea as more palatable than the given tea, and also that the choice effect would be obtained only when the number of options was nine, but not three or twelve. The result supported the hypothesis: the choice effect score for the 9-option condition was significantly positive and the highest, but the scores for the 3-option or the 12-option conditions were not significant. This result was consistent with previous findings [20,22,23] that consumers’ satisfaction with their choice was higher when the number of options was optimal (e.g., eight to ten), but not when the number of options was too small (e.g., two to four) or too large (e.g., more than twelve). It is indicated that, when the number of options is optimal, satisfaction with the choice is increased, and that increased satisfaction with the choice is misattributed to the palatability of the chosen food or beverage. In other words, the choice effect on palatability perception occurs when the number of options is optimal.

It has been suggested that consumers’ satisfaction with their choices increases when the sense of self-determination of their choice is strong, and also that the sense of self-determination increases as a function of the perceived variety of the assortment [21]. Thus, we expected that the perceived variety and the sense of self-determination would be the greatest for the 9-option condition, where the choice effect would be (and was) found. However, in this experiment, the perceived variety and the sense of self-determination were not significantly different among the conditions. This result suggests that further studies are needed to understand the choice effect found in this experiment.

## 4. General Discussion

A growing body of research has shown that our hedonic responses to foods and beverages are not simply determined by their physical and chemical properties [1,2,3,4,5,6,7,8,9,10]. To fully understand consumers’ food behaviors, it is clearly important to investigate the factors that affect the hedonic responses to foods and beverages. Thus, the present study aimed to demonstrate and examine an effect of contextual factors, the choice effect, on the palatability perception of foods and beverages. Experiment 1 failed to clearly demonstrate the choice effect on the palatability perception of curry, but another hypothesis was derived from the results. Experiment 2 examined the hypothesis and successfully demonstrated the choice effect on the palatability perception of tea. 

Experiment 1 aimed to demonstrate the choice effect on the palatability perception of a chosen curry and examine its relationship with expectation. The result showed that the expected palatability rating was high for the high-given condition where the participants believed that they were tasting a curry highly ranked by themselves. However, the perceived palatability ratings were not different among the experimental conditions. Moreover, contrary to previous studies suggesting a strong link between expectation and the actual experience of foods and beverages [17,18,19], there were no significant correlations between the expected and perceived palatability ratings. These results indicate that expectation played little or no role in the choice effect on palatability perception.

The result of Experiment 1 also showed a tendency that the perceived palatability rating was high for the mid-choice condition where participants believed that they were tasting a curry they had chosen from the medium-ranked curries, but it was not significant or consistent. We speculated that the key of the choice effect was the exercise of choice itself, but the choice effect was attenuated since participants might have perceived the number of options as three rather than nine, which is thought of as being too small to increase participants’ satisfaction with their choice [20,22,23].

Experiment 2 examined this hypothesis by manipulating and controlling the number of options. The results showed that when the number of options was nine, but not three or twelve, participants perceived the chosen tea as more palatable than the given tea. This result is surprising since the two cups of tea (ostensibly served as the chosen tea and the given tea) were identical. Therefore, it is indicated that when choice is exercised from an optimal number of options, the choice effect on palatability perception does occur. To our knowledge, this is the first demonstration of the choice effect on the palatability perception of beverages.

Consistent with the suggestion from previous studies [20,22,23], the present study showed that nine options were optimal for the choice effect on palatability perception. How can we understand the optimal number of options for the choice effect? One possibility is as follows. In Experiment 2, for instance, as a function of the number of options the sense of wanted choice (“I selected this particular tea because I wanted to”) tended to increase and the sense of no choice (“I selected this particular tea because I had no choice”) tended to decrease. This indicates that increasing the number of options is essentially beneficial since it increases the sense of self-determination, which is thought to result in greater satisfaction with a choice [21]. However, once the number of options exceeds the processing capacity of the working memory (i.e., seven plus or minus two chunks of information) [26], people are often confused and overwhelmed [15,27]. Those negative experiences might then impair satisfaction with a choice, which in turn impairs the palatability perception of the chosen food or beverage. This can be why the choice effect was obtained in the 9-option condition but not in the 12-option condition.

The present study focused on the palatability perception of foods and beverages. Palatability (or pleasantness) perception is thought to be hedonic or emotional experiences of the sensory aspects of foods, and thus distinct from food preference or liking, which is thought to include more cognitive and behavioral components [2,11,12]. An intriguing research question is whether the choice effect has impacts not only on short-term hedonic experience of foods (found in the present study) but also on consumers’ long-term food preference. The question has not been examined yet, but the choice effect can have a long-term impact. For instance, a previous study [20] examining the choice effect in the context of a placebo treatment has demonstrated that participants who chose a treatment from an optimal number of options were more satisfied with their treatment compared to their counterparts who chose from too small or large number of options. Intriguingly, two weeks after the initial treatment session, participants who were afforded an optimal number of options reported fewer symptoms (i.e., placebo effect) than their counterparts; the choice effect has a long-term impact. Long-term impacts of the choice effect on food preference should be examined in future by recording temporal changes of the participants’ preference for the chosen food product.

One of the practical implications of the present study for the domain of food business is that marketers should carefully manage the number of available options on their food/drink menus, in stores, or web-sites. As shown in a previous study, large options indeed attract customers’ attention at first, but customers in such a situation often hesitate to decide which one to buy, and end up buying nothing [15]. Our findings further indicate that choice from too large (or too small) number of options has no benefit on costumers’ palatability perception, but that choosing from an optimal and manageable number of options makes costumers perceive the food or beverage as more palatable. Intriguingly, it has been suggested that the optimal number of options varies as a function of variables, such as option set complexity, decision difficulty, preference certainty, and decision goal [27,28]. Therefore, by considering these variables, marketers should carefully determine how many options of a certain product category is optimal for their customers, and should try to keep the number of options optimal and manageable.

Another implication is that affording customer multiple opportunities of choice might greatly increase the palatability perception of a food or beverage. For instance, a coffee shop where customers can customize their choice in a number of aspects, such as the type of beans, origin, roasting, grinding and so forth. Each aspect might consist of a manageable number of options (e.g., eight types of roasting: Cinnamon, Light, Medium, High, City, Full City, French, and Italian). In this way, customers are afforded multiple opportunities of choice from a manageable number of options. Even if the outcome of the customization does not perfectly match the customers’ original preference, this system might greatly increase customers’ satisfaction with their choice, and can thus increase the palatability perception of their chosen coffee. It is meaningful for both researchers and marketers to examine this possibility in the future.

It should be noted that the only two types of food products (i.e., curry and tea) were used in the experiments. Furthermore, in Experiment 2 the only one tea product was sampled by the participants, which was aimed to control the choice outcome and examine the choice effect more directly [20,21]. We do not think that these facts significantly limit our findings, but it is still unclear whether the choice effect observed in the present study is restricted to the particular food product used in the experiment. This point should be clarified by further research examining the choice effect where a variety of food products are presented and sampled. By establishing such empirical evidences, we can strengthen the hypothesis that the choice effect on palatability perception can basically be applicable to any other foods and beverages.

## 5. Conclusions

The present study aimed to demonstrate the choice effect on the palatability perception of foods and beverages, using curry and tea as model products. The results indicate that the exercise of choice from an optimal number of options, even when the choice is ostensible, makes participants perceive their chosen curry and tea as being more palatable. Although the psychological mechanism underlying the choice effect still needs to be elucidated further, we believe that the present study sheds light on a new aspect of the effect of contextual factors on our hedonic responses to foods and beverages.

## Figures and Tables

**Figure 1 foods-08-00145-f001:**
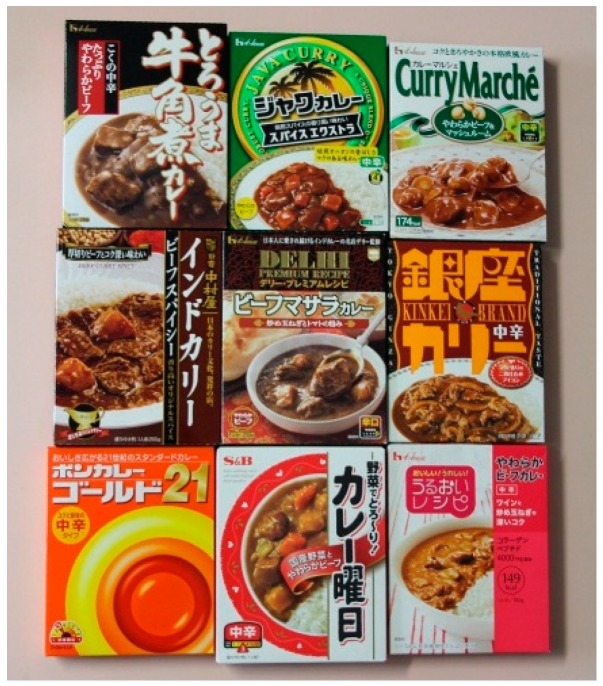
Nine kinds of ready-to-serve curries presented as options in Experiment 1.

**Figure 2 foods-08-00145-f002:**
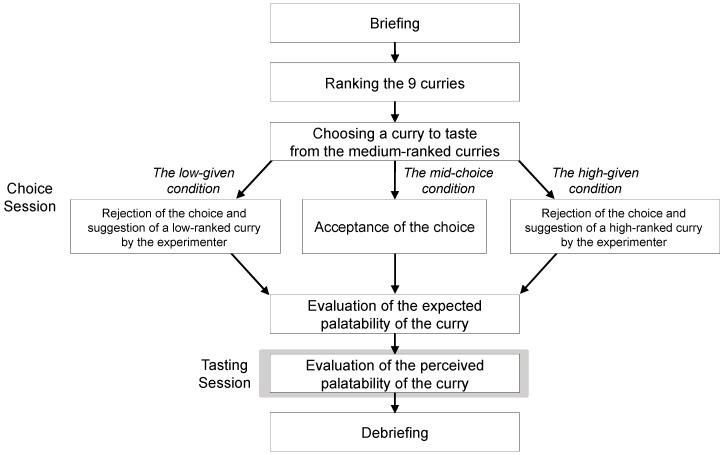
Summary of the experimental procedure.

**Figure 3 foods-08-00145-f003:**
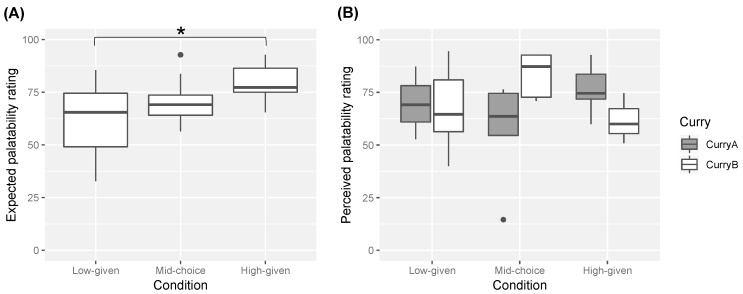
Boxplots of (**A**) the expected palatability rating before tasting and (**B**) the perceived palatability rating during tasting. * represents a significant difference (*p* < 0.05). The black spots show outliers.

**Figure 4 foods-08-00145-f004:**
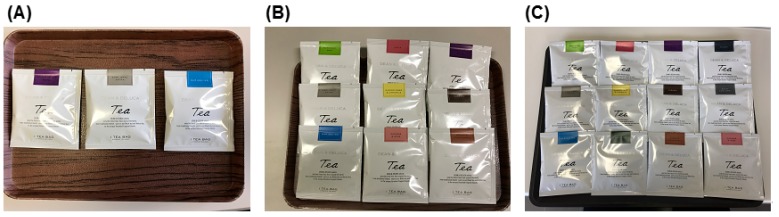
Presented options in (**A**) the 3-option condition, (**B**) the 9-option condition, and (**C**) the 12-option condition in Experiment 2.

**Figure 5 foods-08-00145-f005:**
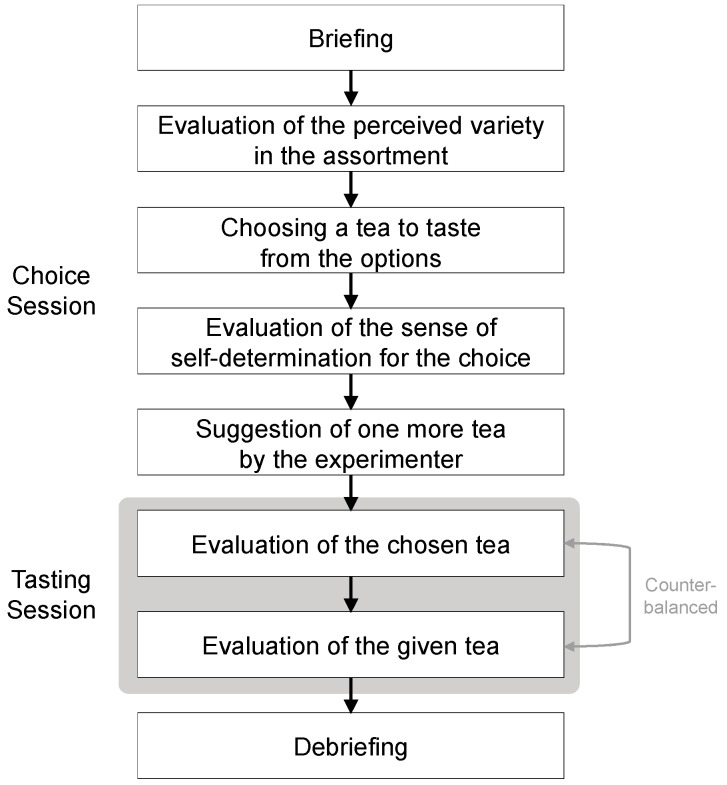
Summary of the experimental procedure. Apart from the number of options presented in the choice session, the procedure was the same among the experimental conditions.

**Figure 6 foods-08-00145-f006:**
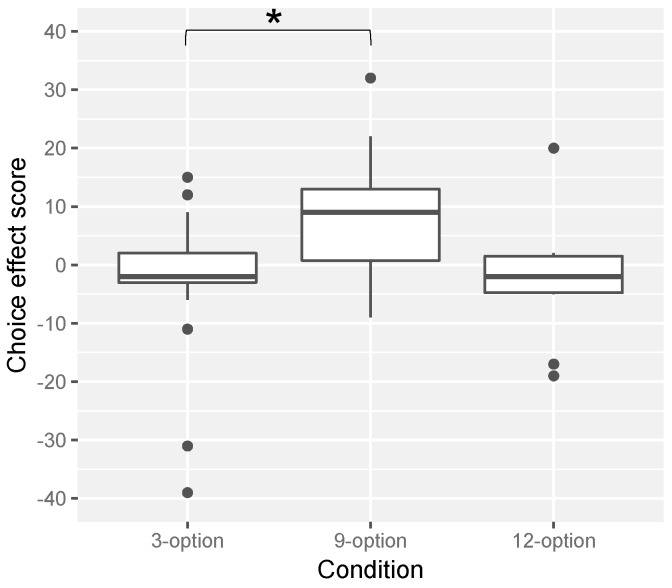
Boxplot of the choice effect score. * represents a significant difference (*p* < 0.05). The black spots show outliers.

**Figure 7 foods-08-00145-f007:**
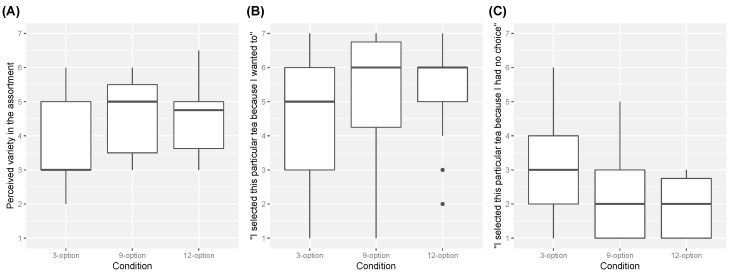
Boxplots of (**A**) the perceived variety in the assortment, (**B**) the score for the item “I selected this particular tea because I wanted to,” (**C**) and the score for the item “I selected this particular tea because I had no choice.” The black spots show outliers.

**Table 1 foods-08-00145-t001:** Spearman’s rank correlation coefficients between the expected palatability rating and the perceived palatability ratings.

Global	The Low-Given	The Mid-Choice	The High-Given
0.01 (*p* = 0.95)	0.13 (*p* = 0.71)	−0.28 (*p* = 0.43)	−0.30 (*p* = 0.40)

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
