# Peer review of "Choosing from an Optimal Number of Options Makes Curry and Tea More Palatable"

_foods, 2019, doi:10.3390/foods8050145_

Round 1

Reviewer 1 Report

The manuscript “Choosing from an Optimal Number of Options Makes Foods and Beverages More Palatable” conducted some experiments to test the influence of the choice effect on consumer palatability. This topic is of interest and the study shows a good set of results.

The main points need to be clarified

L140: Steel-Dwass test was used to test significance in ratings.

Please give some details of this test (what is the effects, fixed or random effects…)

L120-124: participants were asked to expect the flavor and evaluate the palatability based on this.

What are the flavors and how these flavors relate to expected palatability? Without this information, this is difficult to explain about the differences in expected palatability. Very often, to investigate the relation between expected and perceived ratings, the stimuli for expected ratings are images, verbal descriptions…; and the ones for perceived ratings are actual products. In experiment 1, however, stimuli were not clearly introduced.  Please give further explanation on this point.

L250-255, L256-264: two indices (perceived variety, sense of self-determination) were calculated.

For perceived variety, why both two items “How different…” and “How similar…” need to be used. These are two different aspects, and it is not logical to calculate the average. Perhaps, authors ‘d like to validate the scores for “How difference…”, but authors could use, for example, “not difference” instead of “similar”. Similarly, for the sense of self-determination, there is the same issue when using both “I wanted to…” and “I have no choice…”. Please explain further.

L270-272: one product was used to evaluate the perceived palatability.

Although Fig. 6 points out that the palatability in the condition 9-optiont was significant difference as compared to other conditions. However, it is quite difficult to make conclusion regarding the palatability with one product only. It is better if a range of products is tested, and compare the difference in the palatability patterns. Please give some comments on this.

L278-279: participants with the identical scores of chosen and given ratings were excluded.

It is not clear why the participants were eliminated. Please explain.

Author Response

Comment 1

L140: Steel-Dwass test was used to test significance in ratings.

Please give some details of this test (what is the effects, fixed or random effects...)

Steel-Dwass test is a multiple comparison testing based on pairwise ranking, generally known as a non-parametric equivalent of Tukey-Kramer test. In this study, we had had hypotheses about the patterns of difference among the experimental conditions, thus the test was used as a priori multiple comparison test, not as post hoc test following ANOVA or Kruskal-Wallis test. Therefore, we did not set main, fixed or random effects in this analysis. We have added the following sentences explaining this point in the Data Analysis subsection:

To assess the differences among the experimental conditions, Steel-Dwass test, known as a non-parametric equivalent of Tukey-Kramer test, was used as a priori multiple comparison of the expected palatability rating and the perceived palatability rating (the function pSDCFlig in the package NSM3 in R). (L167-170)

Comment 2

L120-124: participants were asked to expect the flavor and evaluate the palatability based on this.

What are the flavors and how these flavors relate to expected palatability? Without this information, this is difficult to explain about the differences in expected palatability. Very often, to investigate the relation between expected and perceived ratings, the stimuli for expected ratings are images, verbal descriptions...; and the ones for perceived ratings are actual products. In experiment 1, however, stimuli were not clearly introduced. Please give further explanation on this point.

We apologize for our insufficient explanation. In actuality, in Experiment 1, we presented the participants with a package of the curry they were going to taste, and then asked the participants to expect its flavor and evaluate how palatable the curry seemed. We have added this description in the Procedure subsection:

During the preparation, participants were presented with the package of the curry they were going to taste, and asked to expect its flavor and to evaluate how palatable the curry seemed (i.e., the expected palatability) on a 55-mm visual analog scale (VAS). (L144-147)

Comment 3

L250-255, L256-264: two indices (perceived variety, sense of self- determination) were calculated.

For perceived variety, why both two items “How different...” and “How similar...” need to be used. These are two different aspects, and it is not logical to calculate the average. Perhaps, authors ‘d like to validate the scores for “How difference...”, but authors could use, for example, “not difference” instead of “similar”. Similarly, for the sense of self-determination, there is the same issue when using both “I wanted to...” and “I have no choice...”. Please explain further.

This method relied on a previous study which demonstrated a relationship among the perceived variety, the sense of self-determination, and satisfaction of choice (Mogliner et al., 2008 [21]). Based on the previous study, we expected that in our study the two different items for each rating were measuring the same concept, but we admitted that our expectation should have been tested. To examine the internal reliability of the two different items for each rating, we calculated Cronbach’s α coefficients. The coefficient for the perceived variety was quite high (α= 0.808), thus we have decided to keep using the average of the scores for the two items (i.e., the scores for “How different…” and the reverse scores for “How similar…”). On the other hand, the coefficient for the sense of self-determination was not high (α= 0.492), thus we have individually analyzed the scores for the two items (i.e., “I wanted to…” and “I had no choice…”) instead of using the average. However, this alteration has not significantly changed the conclusions. We have added the following texts in the Data Analysis subsection:

The perceived variety in the assortment was measured by the two different items. Since internal reliability of the two items was high (Cronbach’s α = 0.808), scores for the first item (“How different are the tea options from each other?”) and reverse scores for the second item (“How similar are the tea options to each other?”) were averaged to create an index of perceived variety.

The sense of self-determination was also measured by the two different items, but internal reliability was not high (Cronbach’s α = 0.492). Therefore, scores for the two items (“I selected this particular tea because I wanted to” and “I selected this particular tea because I had no choice”) were individually analyzed as indices of the sense of self-determination. (L341-348)

Comment 4

L270-272: one product was used to evaluate the perceived palatability.

Although Fig. 6 points out that the palatability in the condition 9-optiont was significant difference as compared to other conditions. However, it is quite difficult to make conclusion regarding the palatability with one product only. It is better if a range of products is tested, and compare the difference in the palatability patterns. Please give some comments on this.

We agree with the reviewer’s concern. We believe that our finding, the choice effect on palatability perception, can be applicable to any other foods and beverages, but it should be tested further. We have added the following paragraph in the General Discussion section:

It should be noted that the only two types of food products (i.e., curry and tea) were used in the experiments. Furthermore, in Experiment 2 the only one tea product was sampled by the participants, which was aimed to control the choice outcome and examine the choice effect more directly [20,21]. We do not think that these facts significantly limit our findings, but it is still unclear whether the choice effect observed in the present study is restricted to the particular food product used in the experiment. This point should be clarified by further research examining the choice effect where a variety of food products are presented and sampled. By establishing such empirical evidences, we can strengthen the hypothesis that the choice effect on palatability perception can basically be applicable to any other foods and beverages. (L508-530)

Comment 5

L278-279: participants with the identical scores of chosen and given ratings were excluded.

It is not clear why the participants were eliminated. Please explain.

We did not exclude the participants with the identical scores of the two ratings. We excluded the participants who reported in the debriefing session that they had noticed the two cups of tea being identical. If the participants had noticed it, they would have also noticed the true purpose of the experiment (i.e., examining the choice effect by comparing the ratings of two identical beverages), which would have altered their response and rating. That’s the reason we excluded the data of the participants. To clarify this point, we have added the following text in the Procedure subsection:

After the tasting session, participants were asked to verbally report anything special they had felt or noticed during the experiment. Participants were then informed the true purpose of the experiment. (L310-312)

We have also changed the following text in the Data Analysis subsection from:

Participants who noticed that the chosen and the given tea were identical were excluded from the following analyses. As a result, one participant in the 3-option condition, three participants in the 9-option condition, and one participant in the 12-option condition were excluded.

To:

Some participants (n = 5) reported that they had noticed the chosen and the given tea being identical. They might have also noticed the true purpose of this experiment, which could have altered their response and rating. Therefore, one participant in the 3-option condition, three participants in the 9-option condition, and one participant in the 12-option condition were excluded from the following analyses. (L331-335)

Reviewer 2 Report

This paper concludes that consumers prefer certain foods when they are able to make choices among approximately nine options, as opposed to three or twelve, and that they perceive differences based on this even when none exist.  Research on food palatability often crosses over into psychology, and the authors have provided ample explanations for their results that cover this area of science.  The implications for food companies are also detailed.  This is a complete study that should be widely cited.

Author Response

Comment

This paper concludes that consumers prefer certain foods when they are able to make choices among approximately nine options, as opposed to three or twelve, and that they perceive differences based on this even when none exist.  Research on food palatability often crosses over into psychology, and the authors have provided ample explanations for their results that cover this area of science.  The implications for food companies are also detailed.  This is a complete study that should be widely cited.

We greatly appreciate for the reviewer’s comment. It encourages us a lot.

Reviewer 3 Report

The manuscript seems interesting and well written to me. However, it shows some aspects to be improved. The title for my opinion does not reflect the real content of the paper. You talk about food and benverage, but in the text are described only 2 specific products. In my opinion you can talking about curries products and tea. 

abstract: clarify lines 16-19 .

keywords: I would add tea and curries

Introduction: I would include the definition of palatable and some more references related to the mechanism of consumer choice also in comparison to other food products (not ready to eat) or beverages.

line 34: take off food and beverage is a repetition.

procedures: is it a likert scale?

better explain the experimental phase.Generally, also in the discussion you can improve references. 

the conclusions are limited: they should be extended and include the limits of the work (e.g. only 2 particular products)  

Author Response

<Reviewer 3>

Comment 1

The title for my opinion does not reflect the real content of the paper. You talk about food and beverage, but in the text are described only 2 specific products. In my opinion you can talking about curries products and tea.

We appreciate this suggestion. As suggested, we have changed the title from:

Choosing from an Optimal Number of Options Makes Foods and Beverages More Palatable

 To:

Choosing from an Optimal Number of Options Makes Curry and Tea More Palatable (L2-3)

Comment 2

abstract: clarify lines 16-19.

As suggested, we have clarified that part of the abstract. We have changed the following text from:

Although Experiment 1 failed to demonstrate significant results, Experiment 2 demonstrated that participants perceived a tea chosen by themselves as more palatable than another tea chosen by the experimenter, regardless of the fact that the two cups of tea were identical.

To:

Experiment 1 failed to demonstrate significant differences among palatability ratings for a curry chosen by participants and for a curry chosen by the experimenter. However, Experiment 2 demonstrated that participants perceived a tea chosen by themselves as more palatable than another tea chosen by the experimenter, regardless of the fact that the two cups of tea were identical. (L15-17)

Comment 3

keywords: I would add tea and curries

As suggested, we have added “curry” and “tea” in the keyword section (L24).

Comment 4

Introduction: I would include the definition of palatable and some more references related to the mechanism of consumer choice also in comparison to other food products (not ready to eat) or beverages.

Following the previous studies (Prescott, 2012 [11]; Sakai et al., 2001 [2]; Stevenson, 2009 [12]), we defined “palatability” (or “pleasantness”) as hedonic experience of the sensory aspects of foods and beverages, and defined “preference” (or “liking”) as cognitive judgement or resultant selection behavior (L34-39). We have added the following text in the Introduction section:

When the hedonic responses to foods and beverages are investigated by researchers, several different aspects, such as palatability, pleasantness, liking, or preference are discussed. Generally, palatability and pleasantness are thought to be hedonic experience of the sensory aspects of foods and beverages, while liking and preference are thought to be cognitive judgement or resultant selection behavior [2,11,12]. The present study focused on the palatability of foods and beverages, and aimed to examine the effect of choice on palatability perception. (L34-39)

Following this change, we have added some references [11,12]:

11. Prescott, J. Chemosensory learning and flavor: Perception, preference and intake. Physiol. Behav. 2012, 107, 553559.

12. Stevenson, R.J. The Psychology of Flavor; Oxford University Press: Oxford, UK, 2009.

The reviewer also suggested that we should review some studies investigating the mechanism of consumer choice. Of course, we admit that such studies are quite important and worth to be noted, but our research aim in the present study is not to investigate the mechanism of consumer food choice, rather to investigate the effect of choice on palatability perception of foods; the mechanism of consumer choice itself is beyond our scope. Therefore, we have decided not to include such references.

Comment 5

line 34: take off food and beverage is a repetition.

As suggested, we have removed “foods and beverages” (L33).

Comment 6

procedures: is it a likert scale?

We used both Visual Analogue Scales (VAS) and 7-point Likert scales. To clarify this point, we have added the word “Likert scale” where it was described in the Procedure subsection:

This evaluation was conducted, based on previous studies [21,24], by asking participants “How different are the tea options from each other?” and “How similar are the tea options to each other? (reversed item)” on a 7-point Likert scale (1 = not at all, 7 = very much). (L290-293)

And:

This evaluation was conducted, based on previous studies [21,25], by asking participants to state how true the following statements were for them on a 7-point Likert scale (1 = not at all true, 7 = very true). (L296-298)

Comment 7

better explain the experimental phase.

We are afraid that this comment is a bit unclear. However, following the reviewer 1’s and the reviewer 3’s relevant comments, we have revised the experimental phases. We hope the revisions sufficiently respond to this comment.

Comment 8

Generally, also in the discussion you can improve references.

Following the change in the Introduction section (i.e., definition of “palatability,” see also the response to Comment 4), we have added a discussion about a possible impact of the choice effect on long-term food preference, and also suggested a future direction of research. We have added the following text in the General Discussion section:

The present study focused on the palatability perception of foods and beverages. Palatability (or pleasantness) perception is thought to be hedonic or emotional experiences of the sensory aspects of foods, and thus distinct from food preference or liking, which is thought to include more cognitive and behavioral components [2,11,12]. An intriguing research question is whether the choice effect has impacts not only on short-term hedonic experience of foods (found in the present study) but also on consumers’ long-term food preference. The question has not been examined yet, but the choice effect can have a long-term impact. For instance, a previous study [20] examining the choice effect in the context of a placebo treatment has demonstrated that participants who chose a treatment from an optimal number of options were more satisfied with their treatment compared to their counterparts who chose from too small or large number of options. Intriguingly, two weeks after the initial treatment session, participants who were afforded an optimal number of options reported fewer symptoms (i.e., placebo effect) than their counterparts; the choice effect has a long-term impact. Long-term impacts of the choice effect on food preference should be examined in future by recording temporal changes of the participants’ preference for the chosen food product. (L472-485)

Comment 9

The conclusions are limited: they should be extended and include the limits of the work (e.g. only 2 particular products)

We believe that our finding is not restricted to curry or tea only, but its generality should be tested further. To  discuss this point as a limitation, we have added the following text in the General Discussion section:

It should be noted that the only two types of food products (i.e., curry and tea) were used in the experiments. Furthermore, in Experiment 2 the only one tea product was sampled by the participants, which was aimed to control the choice outcome and examine the choice effect more directly [20,21]. We do not think that these facts significantly limit our findings, but it is still unclear whether the choice effect observed in the present study is restricted to the particular food product used in the experiment. This point should be clarified by further research examining the choice effect where a variety of food products are presented and sampled. By establishing such empirical evidences, we can strengthen the hypothesis that the choice effect on palatability perception can basically be applicable to any other foods and beverages. (L508-530)

In the Conclusion section, to clarify the limitation, we have added a sentence “using curry and tea as model products”:

The present study aimed to demonstrate the choice effect on the palatability perception of foods and beverages, using curry and tea as model products. (L532-533)

Considering the limitation, we have also changed the following text from:

The results indicate that the exercise of choice from an optimal number of options, even when the choice is ostensible, makes people perceive their chosen foods and beverages as being more palatable.

To:

The results indicate that the exercise of choice from an optimal number of options, even when the choice is ostensible, makes participants perceive their chosen curry and tea as being more palatable. (L533-535)

Round 2

Reviewer 1 Report

The revision has improved the quality of manuscript.

It is well interpreted now.